# Discovering Low-Precision Networks Close to Full-Precision Networks for Efficient Embedded Inference

## Abstract

To realize the promise of ubiquitous embedded deep network inference, it is essential to seek limits of energy and area efficiency. To this end, low-precision networks offer tremendous promise because both energy and area scale down quadratically with the reduction in precision. Here, for the first time, we demonstrate ResNet-18, ResNet-34, ResNet-50, ResNet-152, Inception-v3, densenet-161, and VGG-16bn networks on the ImageNet classification benchmark that, at 8-bit precision exceed the accuracy of the full-precision baseline networks after one epoch of finetuning, thereby leveraging the availability of pretrained models. We also demonstrate ResNet-18, ResNet-34, and ResNet-50 4-bit models that match the accuracy of the full-precision baseline networks – the highest scores to date. Surprisingly, the weights of the low-precision networks are very close (in cosine similarity) to the weights of the corresponding baseline networks, making training from scratch unnecessary.

We find that gradient noise due to quantization during training increases with reduced precision, and seek ways to overcome this noise. The number of iterations required by stochastic gradient descent to achieve a given training error is related to the square of (a) the distance of the initial solution from the final plus (b) the maximum variance of the gradient estimates. By drawing inspiration from this observation, we (a) reduce solution distance by starting with pretrained fp32 precision baseline networks and fine-tuning, and (b) combat noise introduced by quantizing weights and activations during training, by using larger batches along with matched learning rate annealing. Sensitivity analysis indicates that these techniques, coupled with proper activation function range calibration, offer a promising heuristic to discover low-precision networks, if they exist, close to fp32 precision baseline networks.

## 1 Introduction

### 1.1 Problem Statement

To harness the power of deep convolutional networks in embedded and large-scale application domains requires energy-efficient implementation, leading to great interest in low-precision networks suitable for deployment with low-precision hardware accelerators. Consequently there have been a flurry of methods for quantizing both the weights and activations of these networks (Jacob et al., 2017; Courbariaux et al., 2015; Polino et al., 2018; Xu et al., 2018; Baskin et al., 2018; Mishra et al., 2017; Choi et al., 2018). A common perception is that 8-bit networks offer the promise of decreased computational complexity with little loss in accuracy, without any need to retrain. However, the published accuracies are typically lower for the quantized networks than for the corresponding full-precision net (Migacz, 2017). Even training 8-bit networks from scratch fails to close this gap (Jacob et al., 2017) (See Table 1). The situation is even worse for 4-bit precision. For the ImageNet classification benchmark, only one method has been able to match the accuracy of the corresponding full-precision network when quantizing both the weights and activations at the 4-bit level (Zhuang et al., 2018). The complexity and ad-hoc nature of prior methods motivates the search for a simpler technique that can be consistently shown to match or exceed full-precision baseline networks.

Table 1: **Fine-tuning After Quantization (FAQ) exceeds or matches the accuracy of the fp32 baseline networks on the Imagenet benchmark for both 8 and 4 bits on representative state-of-the-art network architectures, and outperforms all comparable quantization methods in all but one instance.** Baselines are popular architectures He et al. (2016); Huang et al. (2017); Szegedy et al. (2016); Simonyan & Zisserman (2014) from the PyTorch model zoo. Other results reported in the literature are shown for comparison, with methods exceeding or matching their top-1 baseline (which may be different than ours) in bold. Precision is in bits, where w= weight , and a=activation function. Accuracy is reported for the ImageNet classification benchmark. FAQ ResNet-18, 4-bit result shows mean±std for 3 runs. Compared methods: Apprentice (Mishra & Marr, 2017), Distillation (Polino et al., 2018), UNIQ (Baskin et al., 2018), IOA (Jacob et al., 2017), Joint Training (Jung et al., 2018), EL-Net (Zhuang et al., 2018). Since only one epoch was necessary to fine-tune 8-bit models, we were able to study more 8 than 4-bit models.

| Network | Method | Precision (w,a) | Accuracy (% top-1) | Accuracy (% top-5) |
|---|---|---|---|---|
| ResNet-18 | baseline | 32,32 | 69.76 | 89.08 |
| **ResNet-18** | **Apprentice** | **4,8** | **70.40** | **-** |
| **ResNet-18** | **FAQ (This paper)** | **8,8** | **70.02** | **89.32** |
| **ResNet-18** | **FAQ (This paper)** | **4,4** | **69.78±0.04** | **89.11±0.03** |
| **ResNet-18** | Joint Training | 4,4 | 69.3 | - |
| ResNet-18 | UNIQ | 4,8 | 67.02 | - |
| ResNet-18 | Distillation | 4,32 | 64.20 | - |
| ResNet-34 | baseline | 32,32 | 73.30 | 91.42 |
| **ResNet-34** | **FAQ (This paper)** | **8,8** | **73.71** | **91.63** |
| **ResNet-34** | **FAQ (This paper)** | **4,4** | **73.31** | **91.32** |
| ResNet-34 | UNIQ | 4,32 | 73.1 | - |
| ResNet-34 | Apprentice | 4,8 | 73.1 | - |
| ResNet-34 | UNIQ | 4,8 | 71.09 | - |
| ResNet-50 | baseline | 32,32 | 76.15 | 92.87 |
| **ResNet-50** | **FAQ (This paper)** | **8,8** | **76.52** | **93.09** |
| **ResNet-50** | **FAQ (This paper)** | **4,4** | **76.27** | **92.89** |
| **ResNet-50** | **EL-Net** | **4,4** | **75.9** | **92.4** |
| ResNet-50 | IOA | 8,8 | 74.9 | - |
| ResNet-50 | Apprentice | 4,8 | 74.7 | - |
| ResNet-50 | UNIQ | 4,8 | 73.37 | - |
| ResNet-152 | baseline | 32,32 | 78.31 | 94.06 |
| **ResNet-152** | **FAQ (This paper)** | **8,8** | **78.54** | **94.07** |
| Inception-v3 | baseline | 32,32 | 77.45 | 93.56 |
| **Inception-v3** | **FAQ (This paper)** | **8,8** | **77.60** | **93.59** |
| Inception-v3 | IOA | 8,8 | 74.2 | 92.2 |
| Densenet-161 | baseline | 32,32 | 77.65 | 93.80 |
| **Densenet-161** | **FAQ (This paper)** | **8,8** | **77.84** | **93.91** |
| VGG-16bn | baseline | 32,32 | 73.36 | 91.50 |
| **VGG-16bn** | **FAQ (This paper)** | **8,8** | **73.66** | **91.56** |

## 1.2 CONTRIBUTIONS

Guided by theoretical convergence bounds for stochastic gradient descent (SGD), we propose fine-tuning, training pretrained high-precision networks for low-precision inference, by combating noise during the training process as a method for discovering both 4-bit and 8-bit integer networks. We evaluate the proposed solution on the ImageNet benchmark on a representative set of state-of-the-art networks at 8-bit and 4-bit quantization levels (Table 1). Contributions include the following.

- We demonstrate 8-bit scores on ResNet-18, 34, 50, and 152, Inception-v3, Densenet-161, and VGG-16 exceeding the full-precision scores after just one epoch of fine-tuning.

- We present evidence of 4 bit, fully integer ResNet-18, ResNet-34, and ResNet-50 networks, which match the accuracy of the original full-precision networks on the ImageNet benchmark, settting the new state-of-the-art.

- We present empirical evidence for gradient noise that is introduced by weight quantization. This gradient noise increases with decreasing precision and may account for the difficulty in fine-tuning low-precision networks.

- We demonstrate that progressively larger batches during fine-tuning provides further accuracy improvements.

- We find direct empirical support that, as with 8-bit quantization, near optimal 4-bit quantized solutions exist close to high-precision solutions, making training from scratch unnecessary.

### 1.3 PROPOSED SOLUTION

Our goal is to quantize existing networks to 8 and 4 bits for both weights and activations, without increasing the computational complexity of the network to compensate, e.g. with modifications such as feature expansion, while achieving accuracies that match or exceed the corresponding full-precision networks. For precision below 8 bits, the typical method is to train the model using SGD while enforcing the constraints (Courbariaux et al., 2015). There are at least two problems faced when training low-precision networks: learning in the face of low-precision weights and activations, and capacity limits in the face of these constraints. Assuming that capacity is sufficient and a low-precision solution exists, we wish to solve the first problem, that is, find a way to train low-precision networks to obtain the best possible score subject to capacity limits.

We use low-precision training to optimize quantized networks. We hypothesize that noise introduced by quantizing weights and activations during training is the crux of the problem and is a second source of noise that is similar to gradient noise inherent to stochastic gradient descent. In support of this idea, Polino et al. (2018) showed that unbiased quantization of weights is equivalent to adding Gaussian noise to the activations, in expectation. The problem is then to find ways to overcome this noise. SGD requires [1]

$$k \leq (\sigma^2 + L * \|x_0 - x^*\|_2^2)^2 / \epsilon^2 \qquad (1)$$

iterations to find a $2\epsilon$-approximate optimal value, where $\sigma^2$ is the gradient noise level, $L$ is related to the curvature of the convex function, $x_0$ and $x^*$ are the initial and optimal network parameters, respectively, and $\epsilon$ is the error tolerance (Meka, 2017). This suggests two ways to minimize the final error. First, start closer to the solution, i.e. minimize $x_0 - x^*$. We therefore start with pretrained models for quantization, rather than training from scratch (Zhou et al., 2017; Baskin et al., 2018). Second, minimize $\sigma^2$. To do this, we combine well-known techniques to combat noise: 1) larger batches which reduce the gradient noise proportional to the square root of the batch size (Goodfellow et al., 2016), and 2) learning rate annealing to lower learning rates ($10^{-6}$), effectively averaging over more batches (batch size increases and learning rate decreases are known to behave similarly (Smith et al., 2017)). Additionally, in the case of 4-bit precision, we fine-tune longer – 110 epochs – to achieve better accuracy, according to equation 1. Finally, we use the initial pretrained network to determine the proper ranges for quantizing both the weights and activations. We refer to this technique as Fine-tuning after quantization, or FAQ. We argue that the method of fine-tuning for quantization is the right approach in the sense that it directly optimizes the proper objective function, the final score, rather than proxies which measure distance from the full-precision network parameters (Migacz, 2017).

## 2 BACKGROUND

### 2.1 NETWORK QUANTIZATION

In the quest for training state-of-the-art low-precision networks, there has been a vast diversity in how the precision constraints are imposed as well as in approaches used in their training. Typical variations in applying low-precision constraints include allowing non-uniform quantization of

---

[1] This assumes a convex loss function, a simpler case.

weights and activations (Miyashita et al., 2016; Zhou et al., 2017; Cai et al., 2017) where the discrete dictionary may depend on the data, and stochastic quantization (Polino et al., 2018; Courbariaux et al., 2015). Approaches to training these networks include distillation (Polino et al., 2018), layer-wise quantization and retraining (Xu et al., 2018), introducing noise during training (Baskin et al., 2018), increasing features (Mishra et al., 2017), learning quantization-specific parameters using backpropagation (Choi et al., 2018), fine-tuning (Baskin et al., 2018; Zhuang et al., 2018), using Stochastic Variance-Reduced Gradient instead of SGD (Sa et al., 2018), and relaxation methods resembling annealing (Yin et al., 2018).

With notable exception of a few papers dealing with binary or trinary networks (Courbariaux et al., 2015; Rastegari et al., 2016; Courbariaux & Bengio, 2016)[2], most of the literature on low-precision networks constrain the number of discrete values that the weights and activations assume but otherwise allow them to be floating-point numbers. In addition, low-precision constraints are not necessarily imposed on batch-normalization constants, average-pooling results etc. in these networks. This is in contrast to how 8-bit integer networks are supported by TensorRT as well as Tensor-Flow framework, where all the parameters and activations are quantized to 8-bit fixed-point integers (see for example (Jacob et al., 2017)). Recent attempts (Wu et al., 2018) at training low-precision networks with integer constraints have hinted at the possibility of porting such networks to commercially available hardware for inference[3].

We focus on training networks with both weights and activations constrained to be either 4 bit, or 8-bit fixed-point integers, and restrict all other scalar multiplicative constants (for example, batch-normalization) in the network to be 8-bit integers and additive constants (for example, bias values) to be 32-bit integers.

## 3 LOW-PRECISION FINE-TUNING METHODS

We start with pretrained, high-precision networks from the PyTorch model zoo, quantize, and then fine-tune for a variable number of epochs depending on the precision. We hypothesize that noise is the limiting factor in finding low-precision solutions, and use well-known methods to over come noise in training. Otherwise, we use the techniques of Courbariaux et al. (2015); Esser et al. (2016) to train low-precision networks. Details of this procedure are described next.

### 3.1 FIXED POINT QUANTIZER

The quantizer we use throughout this paper is parametrized by the precision (in number of bits) $b$, and the location of the least significant-bit relative to the radix $l$, and denoted by $Q_{b,l}$. A calibration phase during initialization is used to determine a unique $l$ for each layer of activations, which remains fixed subsequently throughout the fine-tuning. Similarly, each layer's weight tensor as well as other parameter tensors are assigned a unique $l$ and this quantity is determined during each training iteration. The procedures for determining $l$ for activations and other parameters are described in the following subsections. A given scalar $x$ is quantized to a fixed-point integer $\hat{x} = Q_{b,l}(x) = \min(\lfloor x \times 2^{-l} \rfloor, 2^b - 1) \times 2^l$ for unsigned values, and $\hat{x} = \max(\min(\lfloor x \times 2^{-l} \rfloor, 2^{b-1} - 1), -2^{b-1} + 1)) \times 2^l$ for signed values.

Given a desired network precision of either 8 or 4 bits, we quantize all weights and activations to this level. In the 4-bit case, we leave the first and last layer weights at 8 bits and allow full-precision (32-bit fixed point) linear activations in the last, fully-connected layer Courbariaux et al. (2015); Esser et al. (2016). In addition, the input to that last, fully-connected layer is also allowed to be an 8-bit integer as is the common practice in the literature. In such networks containing 4-bit internal layers and 8-bit final layer, the transition from 4-bit to 8-bit is facilitated by the last ReLU activation layer in the network. Every other ReLU layer's output tensor is quantized to a 4-bit integer tensor.

### 3.1.1 QUANTIZING NETWORK PARAMETERS

Given a weight tensor $w$, SGD is used to update $w$ as usual but a fixed-point version is used for inference and gradient calculation (Courbariaux et al., 2015; Esser et al., 2016). The fixed-point

---

[2]Even these networks may have occasional floating point scaling steps between layers.

[3]NVIDIA's recently announced Turing architecture supports 4-bit integer operations, for example.

version is obtained by applying $Q_{b,l}$ element-wise. The quantization parameter $l$ for a given weight tensor is updated during every iteration and computed as follows: We first determine a desired quantization step-size $\Delta$ by first clipping the weight tensor at a constant multiple[4] of its numerically estimated standard-deviation, and then dividing this range into equally-sized bins. Finally, the required constant $l$ is calculated as $l = \lceil \log_2(\Delta) \rceil$. All other parameters, including those used in batch-normalization, use $l = -b/2$.

## 3.2 INITIALIZATION

Network parameters are first initialized from an available pretrained model file (https://pytorch.org/docs/stable/torchvision/models.html). Next, the quantization parameter $l$ for each layer of activation is calibrated using the following procedure: Following Jacob et al. (2017), we use a technique of running several (5) training data batches through the unquantized network to determine the maximum range for uniform quantization. Specifically, for each layer, $y_{max}$ is the maximum across all batches of the 99.99th percentile of the batch of activation tensor of that layer, rounded up to the next even power of two. This percentile level was found to give the best initial validation score for 8-bit layers, while 99.9 was best for layers with 4-bit ReLUs. The estimated $y_{max}$, in turn, determines the quantization parameter $l$ for that tensor. For ReLU layers, the clipped tensor in the range $[0, y_{max}]$ is then quantized using $Q_{b,l}$. Once these activation function parameters are determined for each of the tensors, they are kept fixed during subsequent fine-tuning.

For control experiments which start from random initialization rather than pretrained weights, we did not perform this ReLU calibration step, since initial activation ranges are unlikely to be correct. In these experiments, we set the maximum range of all ReLU activation functions to $y_{max} = 2^{p/2} - 1$, where $p$ is the number of bits of precision.

## 3.3 TRAINING

To train such a quantized network we use the typical procedure of keeping a floating point copy of the weights which are updated with the gradients as in normal SGD, and quantize weights and activations in the forward pass (Courbariaux et al., 2015; Esser et al., 2016), clipping values that fall above the maximum range as described above. We also use the straight through estimator (Bengio et al., 2013) to pass the gradient through the quantization operator.

For fine-tuning pretrained 8-bit networks, since the initial quantization is already within a few percent of the full-precision network in many cases, we find that we need only a single additional epoch of training, with a learning rate of $10^{-4}$ after the initial quantization step, and no other changes are made to the original training parameters during fine-tuning.

However, for 4-bit networks, the initial quantization alone gives poor performance, and matching the performance of the full-precision net requires training for 110 additional epochs using exponential decay of the learning rate such that the learning rate drops from the initial rate of 0.0015 (slightly higher than the final learning rate used to train the pretrained net) to a final value of $10^{-6}$. Accordingly we multiply the learning rate by 0.936 after each epoch for a 110 epoch fine-tuning training run. In addition, for the smallest ResNet 4-bit network, ResNet-18, the weight decay parameter is reduced from $10^{-4}$ used to train ResNet models to $0.5 \times 10^{-4}$ assuming that less regularization is needed with smaller, lower precision networks. The batch size used was 256 split over 2 GPUs. SGD with momentum was used for optimization. Software was implemented using PyTorch.

## 4 EXPERIMENTS

### 4.1 FINE-TUNING MATCHES OR EXCEEDS THE ACCURACY OF THE INITIAL HIGH-PRECISION NETWORK

FAQ trained 8-bit networks outperform all comparable quantization methods in all but one instance and exceeded pretrained fp32 network accuracy with only one epoch of training following quantization for all networks explored (Table 1). Immediately following quantization, network accuracy was

---

[4]The constant, in general, depends on the precision. We used a constant of 4.12 for all our 4-bit experiments.

Table 2: **Sensitivity experiments indicate that longer training duration, initialization from a pretrained model, larger batch size, lower weight decay, and initial activation calibration all contribute to improved accuracy when training the 4-bit ResNet-18 network, while the exact learning rate decay schedule contributed the least.** The standard parameters are on row 1. Each subsequent row shows the parameters and score for one experiment with changed parameters in bold. * Note that to keep the number of weight updates approximately the same, the number of epochs was inceased, since larger batches result in fewer updates per epoch.

| Epochs | Pre-trained | Batch size | Learning rate schedule | Weight decay | Activation calibration | Accuracy (% top-1) | Change |
|--------|-------------|------------|------------------------|--------------|------------------------|--------------------|--------|
| 110 | Yes | 256 | exp. | 0.00005 | Yes | 69.82 | - |
| **60** | Yes | **400** | exp. | 0.00005 | Yes | 69.40 | -0.22 |
| 110 | **No** | 256 | exp. | 0.00005 | Yes | 69.24 | -0.58 |
| 165* | Yes | **256-2048** | exp. | 0.00005 | Yes | 69.96 | +0.14 |
| 110 | Yes | 256 | **step** | 0.00005 | Yes | 69.90 | +0.08 |
| 110 | Yes | 256 | exp. | **0.0001** | Yes | 69.59 | -0.23 |
| 110 | Yes | 256 | exp. | 0.00005 | **No** | 69.19 | -0.63 |

nearly at the level of the pretrained networks (data not shown) with one exception, Inception-v3, which started at 72.34% top-1. Since the networks started close to a good solution, they did not require extensive fine-tuning to return to and surpass pretrained networks.

FAQ trained 4-bit network accuracy exceeds all comparable quantization methods, surpassing the next closest approach by nearly 0.5% for ResNet-18 (Jung et al., 2018), and matched or exceeded pretrained fp32 network accuracy. Four-bit networks required significantly longer fine-tuning – 110 epochs – for the networks trained, ResNet-18, ResNet-34, and ResNet-50. In contrast to the 8-bit cases, immediately following quantization, network accuracy dropped precipitously, requiring significant fine-tuning to match and surpass the pretrained networks.

FAQ trained 4-bit network accuracy is sensitive to several hyperparameters (Table 2). We elaborate on some of these results subsequently.

## 4.2 LONGER TRAINING TIME WAS NECESSARY FOR 4-BIT NETWORKS

For the 4-bit ResNet-18, longer fine-tuning improved accuracy (Table 2), potentially by averaging out gradient noise introduced by discretization (Polino et al., 2018). We explored sensitivity to shortening fine-tuning by repeating the experiment for 30, 60 and 110 epochs, with the same initial and final learning rates in all cases, resulting in top-1 accuracies of 69.30, 69.40, and 69.68 respectively. The hyperparameters were identical, except the batch size was increased from 256 to 400. These results indicate that training longer was necessary.

## 4.3 QUANTIZING A PRETRAINED NETWORK IMPROVES ACCURACY

Initializing networks with a discretized pretrained network followed by fine-tuning improved accuracy compared with training a quantized network from random initialization for the same duration (Table 2), suggesting proximity to a full-precision network enhances low-precision fine-tuning. For a 4-bit network, we explored the contribution of the pretrained network by training two ResNet-18 networks with standard initialization for 110 epochs, one with the previous learning rate decay schedule[5] and the other with a learning rate from Choi et al. (2018), dropping by a factor of 0.1 at epochs 30, 60, 85, and 95, plus an additional drop to $10^{-6}$ at epoch 95 to match the fine-tuning experiments. These two approaches reached top-1 accuracies of 67.14% and 69.24%, respectively – both less than FAQ's accuracy after 30 epochs and more than 0.5% short of FAQ's accuracy after 110 epochs. The one FAQ change that degraded accuracy the most was neglecting to calibrate activation

---

[5]We used a higher initial learning rate of 0.1, equal to that used to train the full-precision net from scratch, with a decay factor of 0.901, such that final learning rate was $10^{-6}$.

ranges for each layer using the pretrained model, which dropped accuracy by 0.63%. This is another possible reason why training 8-bit networks from scratch has not achieved higher scores in the past (Jacob et al., 2017).

### 4.4 Reducing noise with larger batch size improves accuracy

Fine-tuning with increasing batch size improved accuracy (Table 2). For a 4-bit network, we explored the contribution of increasing the batch size with a Resnet-18 network, which increased top-1 validation accuracy to 69.96%. We scheduled batch sizes, starting at 256 and doubled at epochs 55, 150, 160, reaching 2048 as maximum batch size[6], each doubling effecting a $\sqrt{2}$ factor drop in gradient noise, which is proportional to square root of batch size. We used 165 epochs to approximately conserve the number of weight updates as the 110-epochs 256-batch-size case as our focus here is not training faster but reducing gradient noise to improve final accuracy. The result is consistent with the idea that gradient noise limits low-precision training, however we cannot not rule out possible confounding affects of training for more epochs or the effect of larger batches on the effective learning step size.

### 4.5 The exact form of exponential learning rate decay was not critical

Replacing the exponential learning rate decay with a typical step decay which reduced the learning rate from $10^{-3}$ to $10^{-6}$ in 3 steps of 0.1 at epochs 30, 60, and 90, improved results slightly (+0.08). This suggests that FAQ is insensitive to the exact form of exponential decrease in learning rate.

### 4.6 Reducing weight decay improves accuracy for ResNet-18

For the 4-bit ResNet-18 network, increasing weight decay from $0.5 \times 10^4$ to $10^{-4}$, used in the original pretrained network, reduced the validation accuracy by 0.23% (Table 2). The smaller ResNet-18 may lack sufficient capacity to compensate for low-precision weights and activations with the same weight decay. In contrast, for the 4-bit ResNet-34 and 50 networks, best results were obtained with weight decay $10^{-4}$.

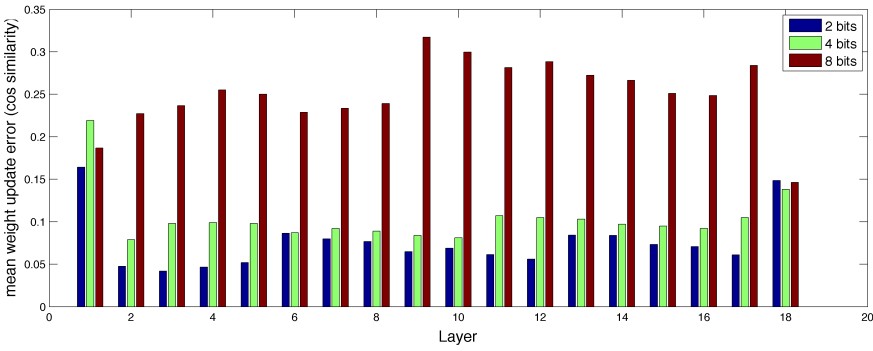

Figure 1: **Quantizing the weights introduces considerable additional noise in the learning process.** Plotted is the cosine of the average angular error between the weight change called for by SGD with momentum, and the actual weight change taken after quantizing. Cosine similarity of 1.0 corresponds to an fp32 network and the absence of discretization-induced gradient noise, i.e. higher is better. This measure is plotted for each layer in ResNet-18 after several hundred iterations in the first epoch of fine-tuning for each of three precisions, 2, 4, and 8 bits for both weights and activations. The first conv layer is layer 1, while the fully connected layer is layer 18. Note that the first and last layer weights are 8 bits in all cases, thus the noise level is similar in all three cases.

---

[6]To simulate batch sizes larger than 256 within memory constraints, we used virtual batches, updating the weights once every $n$ actual batches with the gradient average for effective batch size $256n$.

### 4.7 QUANTIZING WEIGHTS INTRODUCES GRADIENT NOISE

Weight discretization increases gradient noise for 8-, 4-, and 2-bit networks[7]. We define the increase in gradient noise due to weight discretization as the angular difference between the step taken by the learning algorithm, $\delta w$, on the float-point copy at iteration $t-1$, $w_{t-1}$, and the actual step taken due to quantizing the weights, i.e. $Q_{b,l}(w_t) - Q_{b,l}(w_{t-1})$. We measure this angle using cosine similarity (normalized dot-product) between the instantaneous $\delta w$ and an exponential moving average of the actual step directions with smoothing factor $0.9$ (Figure 1). Cosine similarity of $1.0$ corresponds to an fp32 network and the absence of discretization-induced gradient noise. As bit precisions decrease, similarity decreases, signaling higher gradient noise.

These results directly show discretization-induced gradient noise appreciably influences the fine-tuning and training trajectories of quantized networks. The increased noise (decreased similarity) of the 4-bit case compared to the 8-bit case possibly accounts for the difference in fine-tuning times required. Even the 8-bit case is significantly below unity, possibly explaining why training from scratch has not lead to the highest performance (Jacob et al., 2017).

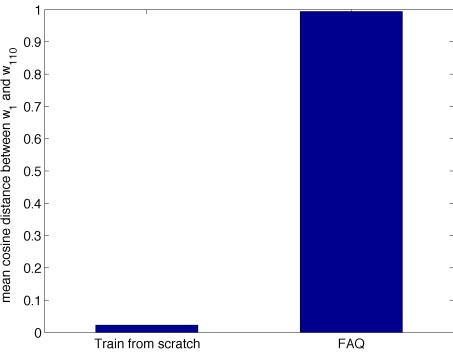

Figure 2: **The ResNet-18 4-bit solution after fine-tuning for 110 epochs was located relatively close to the initial high-precision solution used to initialize the network, indicating that training from scratch is unnecessary.** Plotted is the mean, over all neurons in a ResNet-18 network, of the cosine similarity between the weights at the beginning of training from scratch, and the weights at epoch 110 (left bar). The minimum and maximum similarity measure is 0 and 1, respectively. The similarity between the random initial weights and the final solution is near 0 in this control experiment, indicating that the weights have moved far from the initial condition when training from scratch. The right bar shows the same measure between initial weights taken from the model zoo and the 4-bit solution after 100 epochs of FAQ training. The cosine similarity is close to 1, indicating that the 4-bit solution is close to the initial fp32 solution used for initialization.

### 4.8 THE 4-BIT SOLUTION WAS SIMILAR TO THE HIGH-PRECISION SOLUTION

The weights of the FAQ trained 4-bit network were similar to those in the full-precision pretrained network used for its initialization (Figure 2). We define the network similarity as the cosine similarity between the networks' weights. The average of the cosine similarity between the weights of every corresponding neuron in the two networks is very close to $1$ ($0.994$), indicating that the weight vectors have not moved very far during 110 epochs of fine-tuning and that the 4-bit network exists close to its high-precision counterpart, demonstrating that pretrained initialization strongly influenced the final network. Contrast this with the same measure when training from scratch, where the similarity between the initial weights and final weights is close to 0 ($0.023$). The fact that the 4-bit solution was close to the high-precision solution suggests that training from scratch is unnecessary.

---

[7]2-bit network is used only to demonstrate how discretization-induced gradient noise varies with bit precision.

### 4.9 FAQ GENERALIZES TO CIFAR10

FAQ trained models are as accurate as the full-precision model for ResNet-18 adapted for the CI-FAR10 dataset [8]. The full-precision model was well-trained for 350 epochs, with learning rate 0.1 reduced by a factor of 0.1 at 150 and 250 epochs, momentum=0.9, weight decay=5e-4, batch size=128, and augmentation consisting of random crops from images padded with 4 pixels and randomly flipped horizontally. The baseline test accuracy was 94.65%, while FAQ 8- and 4-bit scores, respectively were 94.65% and 94.63%, evidence that FAQ generalizes to other datasets. 8-bit parameters were the same as that for the Imagenet experiments, except weight decay equaled the baseline, and the number of epochs was extended to 10. The 4-bit score was obtained with the same parameters as for Imagenet (Table 2, row5), except with weight decay equal to baseline and initial learning rate of 0.02 (best result among 0.0015, 0.01, 0.02, and 0.04).

## 5 DISCUSSION

We show here that low-precision quantization followed by fine-tuning, when properly compensating for noise, is sufficient to achieve state of the art performance for networks employing 4- and 8-bit weights and activations. Compared to previous work, our approach offers a major advantage in the 8-bit space, by requiring only a single epoch of post quantization training to consistently exceed high-precision network scores, and a major advantage in the 4-bit space by matching high-precision baseline scores with a simpler approach, exceeding published results on ResNet-18, 34 and 50. We find support for the idea that overcoming noise is the main challenge in successful fine-tuning, given sufficient capacity in a network model: longer training times, exponential learning rate decay, very low final learning rate, and larger batch sizes all seem to contribute to improving the results of fine-tuning. SGD is faced with two sources of noise, one inherent to stochastic sampling, and the other due to quantization noise; these techniques may be reducing only one of the sources, or both, and we have not shown that FAQ is directly reducing quantization noise. Further experiments are warranted.

We believe that the success of fine-tuning and the wide availability of pretrained models marks a major change in how low-precision networks will be trained. We conjecture that within every region containing a local minimum for a high-precision network, there exists a subregion(s) which also contains solutions to the lower precision 4-bit nets, provided that the network has sufficient capacity. The experiments reported herein provide support for this conjecture; if true, FAQ should generalize to any model.

Fine-tuning for quantization has been previously studied. In Zhou et al. (2017), increasingly larger subsets of neurons from a pretrained network are replaced with low-precision neurons and fine-tuned, in stages. The accuracy exceeds the baseline for a range of networks quantized with 5-bit weights and 32-bit activations. Our results here with both fixed-precision weights and activations at either 8 or 4 bits suggest that incremental training may have been unnecessary. In Baskin et al. (2018), fine-tuning is employed along with a non-linear quantization scheme during training (see UNIQ in Table 1). We have shown that low-precision quantization followed by proper fine-tuning, is sufficient to achieve even greater accuracy when quantizing both weights and activations at 4 bits. Finally, using a combination of quantizing weights before activations, progressively lower precisions, fine-tuning, and a new loss function, Zhuang et al. (2018) are the first to show that a 4-bit ResNet network can match the top-1 accuracy of a baseline full-precision network. Our results show that a simpler method can achieve this for ResNet-18, 34, and 50.

Future research includes combining FAQ with other approaches, new training algorithms designed specifically to fight the ill-effects of noise (Baskin et al., 2018) introduced by weight quantizaiton, and extending to quantize 2-bit networks. Training in the 2-bit case will be more challenging given the additional quantization noise (Figure 2), and possible capacity limits with 2-bit quantization.

FAQ is a principled approach to quantization. Ultimately, the goal of quantization is to match or exceed the validation score of a corresponding full-precision network. This work demonstrates that 8-bit and 4-bit quantized networks performing at the level of their high-precision counterparts can be obtained with a straightforward approach, a critical step towards harnessing the energy-efficiency of low-precision hardware.

---

[8]https://github.com/kuangliu/pytorch-cifar/blob/master/models/resnet.py

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
