# OpenReview forum: "Discovering Low-Precision Networks Close to Full-Precision Networks for Efficient Embedded Inference"
_ICLR.cc/2019/Conference_

### Official Review · AnonReviewer3 · 2018-10-27
**A good but narrow contribution in a crowded space**

**Rating:** 6
**Confidence:** 3

**Review:**

This manuscript joins a crowded space of methods for low bit quantization to enable inference on more efficient hardware. In the past, these methods often were limited to 8-bit quantization, or smaller networks, or result in accuracy degradation. This paper is part of a recent crop of methods that achieve full accuracy on ResNet50 with 4-bit weights and activations.

The method in this paper is based around a simple, yet powerful observation: Fine-tuning at low precision introduces noise in the gradient. Using the relationship between noise, batch-size and learning rate, that has recently been receiving a lot of attention in the context of large batch training, they compensate for this added noise by increasing the batch size.

I like the simplicity and effectiveness, and believe that this method will be a useful addition to the toolbox for low-precision inference.

Overall, the paper is well written, and the claims are well supported experimentally. Results are demonstrated on a wide range of networks, including various configurations of ResNet, DenseNet, Inception. It's not clear whether these experiments are from a single run. If they are, with sub 1% differences between methods we are getting close to the run-to-run variability, and it would be preferable to see results averaged across multiple runs.

Ultimately, I am on the fence if this is a sufficient contribution for acceptance. In particular, this paper claims "first evidence ... matching the accuracy of full precision". While this may in a narrow technical sense be the case, PACT https://arxiv.org/abs/1805.06085 also works on ResNet50 without an accuracy drop. While this is not published work, it was rejected at ICLR last year, making it hard to recommend acceptance here. There is also work concurrently submitted to this forum (which I obviously don't expect the authors to cite or take into account, but want to mention for the sake of completeness) such as https://openreview.net/forum?id=HyfyN30qt7 which achieves the same or better results, and does not require 8-bit BN scale factors and 32-bit bias.

This manuscript could be made stronger in multiple ways, e.g. by combining with the recently proposed clipping techniques like Choi et al. (2018), and pushing towards 2 or 3 bit training, or eliminating all larger bit-width parameters to make for easier hardware design.

---

> ### Author Response · Authors · 2018-11-18
> **Response to AnonReviewer3**
>
> Overall response:
>
> Thanks for the kind remarks. The fact that the field is crowded indicates the importance of papers on quantization. We argue that our contribution is important and novel. Regarding the narrowness of the contribution, 1) This new procedure is an important discovery that had not been previously known by those in this field. It is surprising that the complex methods proposed in the past to solve the problem can be beat by this straightforward method (Table 1). 2) Although the method is simpler than others, it is not a trivial discovery; our ablation study showed that a confluence of factors (surprisingly long fine-tuning (for 4-bits), starting from pre-trained networks, larger batches, and initial activation range calibration using the pre-trained net) contribute, explaining why prior fine-tuning attempts were less successful (Table 2). 3) As you point out, this simpler technique will be very useful to practitioners, and its simplicity should lead to widespread use (i.e. citations). It has already been cited 3 times, and we have had inquiries regarding porting the method to Tensorflow. 4) We find that the theoretical justification for our solution grounded in equation 1, as well as our supporting experiments showing gradient noise as a function of quantization coarseness (fig. 1) as well as the proximity of 4 bit to high-precision solutions (fig.2) could help focus future efforts. 5) Demonstrating that a wide range of 8-bit and 4-bit networks can achieve the same or better accuracy as the high precision networks in a standard model zoo is a valuable contribution.  The unpublished PACT paper did not show this (they were -0.4% under the baseline for ResNet50 top1 and 1.2% under on ResNet18). Only one other paper, which we were unaware of but now cite, had shown this for ResNet-50 (see comments by reviewer 2), and only for one resnet, using a more complex training paradigm.
>
> Responses to specific comments:
>
> Regarding the number of trials per experiment:  All results were from 1 run, similar to many of the results taken from the literature and included in Table 1.  We have updated Table 1 to show the mean and standard deviation of 3 runs for Resnet-18 at 4-bits. The average top-1 and top-5 scores exceed the full-precision model zoo network, and the standard deviations are quite small.
>
> We strongly disagree with the PACT comparison. We match state-of-the-art full-precision accuracy for both ResNet-18 and ResNet-50, both top1 and top5, at 4 bits, while PACT did not. They were -1.2% from baseline top-1 on ResNet-18, and their ResNet-50 top1 score at 4 bits was 76.5 with a baseline of 76.9 reported in their Table 7. They used preactivation ResNets so the baseline score is different than ours.  We argue that this discovery signals an important change in the way quantization is done at the 4-and 8-bit level; surprisingly, fine-tuning after activation-calibration is a straightforward technique that consistently matches or exceeds the scores of corresponding full-precision networks in the standard PyTorch model repository; this simplicity should lead to widespread adoption. PACT did not achieve this accuracy with more complex methods.
>
> Finally, the submission you mentioned, NICE, also uses a more complex training methodology: noise injection, progressive quantization of layers, and learned activation quantization ranges. They also use fp32 maximums and minimums when quantizing, less suitable for efficient hardware implementation. This leaves one to wonder if these ad-hoc methods were the key factor. We show that one can consistently match the high-precision network simply with proper training of a pre-trained model after activation-range calibration, a novel and valuable contribution to the field that could help focus future efforts.
>
> We had run a control experiment with PACT, but it did not improve the results, likely due to the fact that we are starting from a pretrained model and therefore the ReLU ranges can be simply measured rather than learned.  We decided not to include the result since it did not improve results and would lengthen the manuscript further.
>
> 2-bit training is left for future work since matching full-precision network accuracy at 2-bits is far less likely; given that 4-bit inference will be possible on NVIDIA hardware, this case is especially relevant.  Finally, there is little additional hardware overhead to make the bias 32-bits as the accumulator for the dotproduct must be higher precision than the weights and activations anyway. Additionally one higher-precision parameter per neuron adds little to the model memory requirements.
>
> The paper provides evidence of an important, novel discovery. Its simplicity compared with prior work is an advantage that may lead to widespread adoption.  These reasons and the state-of-the-art quantization results across the board warrant publication. We hope that this nudges you over to the “publish” side of the fence.

---

### Official Review · AnonReviewer1 · 2018-11-03
**Review comments on “Discovering Low-Precision Networks Close to Full-Precision Networks for Efficient Embedded Inference”**

**Rating:** 4
**Confidence:** 5

**Review:**


Summary:
This paper proposes three methods to improve the performance of the low-precision models. Firstly, to reduce the number of training iterations, the authors propose to do quantization on pre-trained models rather than training from scratch. Secondly, the authors propose to use large batches size and proper learning rate annealing with longer training time to reduce the gradient noise introduced in quantization. Experimental results demonstrate the effectiveness of the proposed methods.

Contributions:
1.	The authors hypothesize that noise introduced by quantization is the limiting factor for training low-precision networks and present empirical evidence to support this hypothesis.
2.	The authors formulate the error as equation (1) and propose two techniques (large batches size and proper learning rate annealing) to minimize the final error.
3.	The authors conduct a series of experiments to demonstrate the effectiveness of the proposed methods.

Cons:
1.	The novelty of this paper is limited. Firstly, fine-tune the pre-trained model is a well-known method in quantization. Secondly, using large batches size and proper learning rate annealing are more like tricks in hyper-parameter tuning rather than a method.

2.	In table 2, the performance of the model in the second row (batch size=400) is worse than the baseline ones (batch size=256). In order to keep the same number of weight updates, the author increases the number of epochs during training, which results in performance improvement. Do large batches size really contribute to performance improvement? Whether the performance gain is due to the large batches size or more sampling data?

3.	The authors claim that large batches size can reduce the gradient noise introduced by quantization. It would be better to show the introduced noise with different batch sizes in figure 1.

4.	This paper is not the first time for ResNet-50 with 4-bit quantization to outperform the full-precision network. EL-Net[1] has trained a 4-bit precision network, which leads to no performance degradation in comparison with its full precision counterpart.

5.	The title in experiments part is too long and confusing. It will be better to keep the short and meaningful title.


References
[1] Zhuang B, Shen C, Tan M, et al. Towards Effective Low-bitwidth Convolutional Neural Networks[J]. 2017.

---

> ### Author Response · Authors · 2018-11-18
> **Response to AnonReviewer1**
>
> Thanks for your feedback which has helped us improve the paper, especially pointing out that we missed reference #1, which is now prominently highlighted.
>
> Response to Con1:
>
> The paper is important and novel for several reasons. 1) This new quantization procedure properly combines previous elements to set the state-of-the-art, an important discovery that had not been previously known: proper fine-tuning of a pretrained model after activation-range calibration is sufficient to match or exceed the results of all of the prior training methodologies and algorithms on a wide range of networks (Table 1). It is surprising that, as you point out, fine-tuning has been employed before, yet always combined with other more complex techniques and yet FAQ outperforms or matches them all. Further, the excellent paper which you point out that exceeded the high-precision score at 4-bits for ResNet-50 used more complex techniques leaving open the question of whether a simpler method could do the same, or whether the complexity was necessary. 2) Although the method is simpler than prior techniques, it is not just hyperparameter tuning tricks; our ablation study showed that a confluence of factors (longer training, starting from pretrained networks, larger batches, and activation range calibration) contribute, explaining why so many prior attempts were less successful (Table 2). 3) The simplicity of the technique will make it attractive to practitioners. It has already been cited 3 times, and we have had inquiries regarding porting the method to Tensorflow. 4) The theoretical justification for our solution grounded in equation 1, as well as our supporting experiments showing gradient noise as a function of quantization coarseness as well as the proximity of 4 bit to high-precision solutions, are novel contributions that may help focus future efforts; perhaps FAQ + other techniques will reduce the training time for 4-bits and transfer to 2-bit networks. 5) Demonstrating that a wide range of 8-bit networks, with only 1 additional training epoch, can achieve the same or better accuracy as the high precision networks, beating the state-of-the-art quantization techniques, in a standard model zoo is a valuable and novel contribution which should not be overlooked (see our reference to very recent paper which could not match high precision nets even at 8-bits!).
>
> Response to Con 2:
>
> Good point.  It is only consistent with the hypothesis that increasing batch size improves the score, but it is not conclusive, for the reasons you mention. Further experiments could tease this apart. We added your point in the results section.
>
> Response to Con 3:
>
> You make a very good point requiring clarification on our part.  We do not prove that larger batches reduce gradient noise due to quantization, but that it helps reduce the overall noise; since SGD convergence is slowed by both noise sources, we try to minimize one of them with larger batches.  We add the following statement to this effect in the discussion to clarify: “SGD is faced with two sources of noise, one inherent to stochastic sampling, and the other due to quantization noise; these techniques may be reducing only one of the sources, or both, and we have not shown that FAQ is directly reducing quantization noise. Further experiments are warranted.”
> Measuring the effect of batch size on quantization noise as you suggest is not so simple, because averaging gradients over larger batches reduces the magnitude of the actual gradients, and thus affect relative quantization errors.
>
> Response to Con 4:
>
> Thank you for pointing out this paper, of which we were unaware.  1) We added a citation to this paper in the discussion and background sections. 2) We added the EL-Net 4-bit result to Table 1. 3) Although this score is lower than our 76.27%, it may be due to differences in data augmentation. 4) FAQ is arguably simpler than EL-Net, and thus is at least an attractive alternative. 5) In addition to 4-bits, we show that FAQ works on a wide range of state-of-the-art networks at 8-bits, which is surprisingly novel (see our reference to Jacob et al, 2017) 6) We remove the primacy claim for 4-bit networks throughout the manuscript to take into account this new reference, and instead state that Huang et al are the first to surpass the full-precision top-1 Imagenet score at 4-bit precision (see the discussion).
>
> Response to Con 5:
>
> We think that the paper is more readable and easier to refer to later when the headings state the main point. However, we shortened some of the subheadings.
>
> We hope that we have addressed your concerns.  Please consider increasing your rating given the improvements due to your comments, and the additional experiments added due to the other reviewers (CIFAR10 experiments and means +/- standard deviations for ResNet-18.)

---

> > ### Comment · AnonReviewer1 · 2018-12-06
> > **further comments**
> >
> > I appreciate the improvements of the resubmitted version by the authors. However,  I still think the novelty of the paper is really limited (essentially some techniques are exploited).  The results in the comparison to EL-Net seems good, but different baselines are actually used. I appreciate good results, but for a ICLR paper, I think new insights are more important.

---

> > > ### Author Response · Authors · 2018-12-07
> > > **Thanks for the feedback**
> > >
> > > Thanks for the additional feedback. I agree that new insights are important.  We contribute several  significant insights.
> > > First, no one had published that, for 8-bit networks, fine-tuning for only 1 epoch AFTER proper activation and weight range calibration (FAQ) exceeds the accuracy of the baseline in all of the models we tried (Table 1), including Densenet161, ResNet152, VGG16bn, and Inception_v3, demonstrating that 8-bit quantization is now efficiently solved for these models.  If everyone had speculated that some method could accomplish this, it remained to be shown (https://arxiv.org/abs/1712.05877), and doing so is an important contribution.
> > >
> > > Second, prior to this work, the best results for 4-bit quantization had combined various clever ad-hoc methods to improve performance.  We contribute the following new key insight: FAQ beats all of these more complex methods by training longer, and finetuning AFTER quantization was critical (Table 2). Further, this holds for a wide range of models including Densenet161, ResNet152, and VGG16bn, indicating that 4-bit quantization is a solved problem. We speculate and show evidence for why this is the case.  Showing that other more complex methods are perhaps helpful, but unnecessary, is an important and novel insight.
> > >
> > > Our rebuttal above points out other contributions as well.  We fail to see why discovering the importance of finetuning only after proper calibration for quantization, and conclusively showing that 8-bit and 4-bit quantization is now a solved problem on a wide range of networks with a model that can be conveniently implemented without ad-hoc algorithm improvements do not constitute key insights.

---

### Official Review · AnonReviewer2 · 2018-11-05
**interesting paper and result is promising, but lack of novelty**

**Rating:** 5
**Confidence:** 4

**Review:**

This paper proposes a fine-tuning scheme for quantized network which can achieve higher accuracy ( in 8bits case) than the original full-precision (32 bits) network. The main finding/motivation of this paper is that in order to make the fine-tuning works, the retrain needs to overcome the gradient noise that is introduced by weight quantization. Therefore, it considers several typical retraining techniques: large batch size training, retraining from full-precision network instead of quantized one from scratch,  lower weight decay.

I think it is an interesting paper, and the result is quite promising. In fact, I have not seen any quantized network that can perform better than the original full-precision network. While in terms of novelty, no new techniques/algorithms are proposed, and it is combing standard strategies used in retrain networks. In addition, I have several questions for this work:

1) It seems that training longer time will benefit the fine-tuning a lot. What if we can also train the original model for some additional amount of training time(like 165 epochs in Table 2), and then quantize this full-precision network without retrain, will the proposed scheme still have better accuracy than this naive way?

2) Will these fine-tuning strategies/findings be generalized to other datasets or other models? In this paper, only results in ImageNet are shown.

3) Can I use these fine-tuning strategies to improve other quantization methods? For example, I could use larger batch size when training for other fine-tuning methods, and will it also make their quantized models better than the original precision model?

4) As mentioned in the paper, the proposed quantized network is used to  speed up the inference time. Some results for inference time using the proposed quantized network will be super interesting.

Overall, the proposed fine-tuning scheme has promising results. My main concern for this paper is its novelty and whether it can be generalized to other models.

---

> ### Author Response · Authors · 2018-11-18
> **Response to AnonReviewer2**
>
> Overall response to lack of novelty.
> Thank you for your careful review. We are gratified that you appreciate the fact that the method yields promising, state-of-the-art results for quantization. We disagree with the novelty assessment for the following reasons.
> 1) This new quantization procedure properly combines previously known elements to achieve state of the art results, an  important discovery that had not been previously known and is therefore novel. Our Table 1 shows that the complex algorithms/methods proposed in the past to solve the problem do not beat our simpler method. Pointing out a simpler solution to an important problem that is better than other complex proposals is a novel contribution. 2) This simpler technique will be very useful to practitioners. It has already been cited 3 times, and we have had inquiries regarding porting the method to Tensorflow. It’s relative simplicity should lead to widespread adoption (and many citations). 3) Although the method is simpler than prior techniques, it is not just combining standard strategies for training; our ablation study showed that a confluence of factors (longer training, starting from pretrained networks, larger batches, and most importantly, activation range calibration from the pretrained model) contribute, explaining why prior attempts were less successful (Table 2). 4) We find the theoretical justification for our solution grounded in equation 1, as well as our supporting experiments showing gradient noise as a function of quantization coarseness (fig.1) as well as the proximity of 4 bit to high-precision solutions (fig. 2), to be novel. 5) Demonstrating that a wide range of 8-bit networks can achieve exceed the accuracy of the high precision networks in a standard model zoo with only 1 epoch of training is also new (see our reference to very recent prior work which could not match high precision nets at 8-bits).
> We feel there is much new and important in this paper. Adding complexity is not the only or the best way to achieve the state-of-the-art quantization results, and theory and experiments in our manuscript suggest why this is so. For these reasons, the paper warrants publication
>
> Responses to questions:
> 1) We ran your experiment on CIFAR10 for the 4-bit case, and FAQ is much better than the suggested control. A Full-precision ResNet18 net was trained with the FAQ learning schedule for 110 epochs, and the top1 accuracy after initial quantization was 93.71%.  Training at 4-bits with FAQ for the same time yields 94.63, nearly a percent higher.
> 2) We added results for CIFAR10 (resnet18 at 4-bits) to the paper. FAQ generalizes to CIFAR10. As far as other models, Table 1 in our paper shows that FAQ generalize to a number of state-of-the-art models.
> 3) We agree and speculate in the discussion that the method could be combined with other improvements that help with training in the presence of noise. FAQ could also be combined with weight pruning, PACT, gradual quantization, etc.
> 4) We have addressed the novelty above. We added an experiment to show that it generalizes to CIFAR10. Table 1 shows that FAQ generalizes to many state-of-the-art models.

---

> > ### Author Response · Authors · 2018-11-25
> > **Additional 4-bit results**
> >
> > In response to your concern about generalization to other models, we have now obtained 77.68% top1 on the 4-bit Densenet161, which is on par with the full-precision baseline from the pytorch model zoo, which is 77.65%.  In addition, 4-bit VGG16bn is 73.70% top1, again exceeding the full-precision baseline which is 73.36.  And 4-bit ResNet-152 also reached the floating point baseline of 78.31% after the 66th epoch. Finally, completing the table for 4-bit networks, inception_v3 finished at 77.33 top-1, -0.12 percent from the full precision score. This indicates that FAQ generalizes well to other models at both 8- and 4-bit precision. Will add these new results to Table 1 after the decision.
> >
> > These results add to the novelty and significance of the paper by providing strong evidence that both 4-bit and 8-bit quantization of deep networks for classification is a solved problem. It is surprising that without many of the complexities of prior methods, a straightforward algorithm that is easily implementable works so well, making it interesting from a theoretical standpoint and at the same time useful to practitioners.

---

### Author Response · Authors · 2018-12-19
**Summary of contributions**

Finding low precision networks for efficient inference was an important open problem in deep learning at both 8- (Jacob et al., 2018) and 4-bit precision (Choi et al, 2018), until now.

A paper published at CVPR earlier this year by researchers at Google (http://openaccess.thecvf.com/content_cvpr_2018/papers/Jacob_Quantization_and_Training_CVPR_2018_paper.pdf) had reported 8-bit integer-only inference scores for several common deep networks that were from 1% to 3% below the full-precision baseline. This is surprising given that 8-bit inference chips from NVidia and Google have existed for several years.

NVidia’s announcement of 4-bit network support in future GPUs makes 4-bit inference timely. The first hints that 4-bit weights and activations might be sufficient for classification came this year from two sources. A group from IBM developed a method, PACT, which achieved close to baseline performance when training networks (ResNet-18 and ResNet-50) from scratch by introducing an algorithm which learned the proper clipping points for the ReLU activation functions used in most deep networks (https://arxiv.org/abs/1805.06085). The top-1 scores were within about 0.5% of the baseline full-precision scores.  Another  paper published at CVPR earlier this year was the first report of 4-bit inference on any network (AlexNet and ResNet-50) which could match the baseline accuracy (http://openaccess.thecvf.com/content_cvpr_2018/papers/Zhuang_Towards_Effective_Low-Bitwidth_CVPR_2018_paper.pdf). Their algorithm, EL-Net, used a combination of techniques including fine-tuning from a pre-trained model, gradually lowering the precision during training, two-stage training in which the weights are quantized prior to activations, and a new loss function requiring the evaluation of a full-precision model along with the quantized model. It remained to be seen whether these techniques were required to reach the accuracy of full-precision networks, or a simpler approach would work. In addition, would 4-bits suffice for shallower or deeper ResNets or more complex nets beyond ResNets.

The present paper shows that 4-bits suffice for classification across a wide range of networks using a much simpler approach called FAQ.  We report matching the reported accuracy of full-precision state-of-the-art deep networks at 4-bits (ResNet-18, -34, -50, and -152, DenseNet-161, and VGG16), demonstrating that finding 4-bit precision networks is a solved problem. The algorithm first quantizes a pre-trained, high-precision model to low-precision by finding proper clipping points for ReLU activation functions and weights which maximize the initial score. The model is then fine-tuned by training longer, 110 epochs for 4-bit solutions, and only 1 epoch for 8-bit solutions. It is shown that starting from a pre-trained model and quantizing properly, and for 4-bit networks, fine-tuning for 110 epochs, were all necessary to match the baseline accuracy for a wide-range of networks. Starting from pre-trained models made the PACT technique of learning the proper activation ranges unnecessary, while simply fine-tuning for much more than the 30 epochs employed by EL-Net obviated the need for additional algorithmic complexity. It is somewhat surprising that without many of the complexities of prior methods, a straightforward algorithm that is easily implementable succeeds, making it interesting from a learning standpoint and at the same time useful to practitioners.

---

### Meta-Review · Area_Chair1 · 2018-12-17
**lack novelty**

**Confidence:** 5
**Recommendation:** Reject

**Metareview:**

This paper proposes methods to improve the performance of the low-precision neural networks. The reviewers raised concern about lack of novelty. Due to insufficient technical contribution, recommend for rejection.